# A Comprehensive Survey on the Expediated Anti-COVID-19 Options Enabled by Metal Complexes—Tasks and Trials

**DOI:** 10.3390/molecules28083354

**Published:** 2023-04-10

**Authors:** Judy Gopal, Manikandan Muthu, Iyyakkannu Sivanesan

**Affiliations:** 1Department of Research and Innovation, Saveetha School of Engineering, Saveetha Institute of Medical and Technical Sciences (SIMATS), Chennai 602105, Tamil Nadu, India; 2Department of Bioresources and Food Science, Institute of Natural Science and Agriculture, Konkuk University, Gwangjin-gu, Seoul 05029, Republic of Korea

**Keywords:** metal complexes, cisplantin, antiviral, COVID-19, mechanism, nanoparticles

## Abstract

Contemporary pharmacology dating back to the late 19th/early 20th centuries has benefitted largely from the incorporation of metal complexes. Various biological attributes have been successfully realized using metal/metal complex-based drugs. Among anticancer, antimicrobial, and antiviral applications, anticancer applications have extracted the maximum benefit from the metal complex, Cisplatin. The following review has compiled the various antiviral benefits harnessed through inputs from metal complexes. As a result of exploiting the pharmacological aspects of metal complexes, the anti-COVID-19 deliverables have been summarized. The challenges ahead, the gaps in this research area, the need to improvise incorporating nanoaspects in metal complexes, and the need to test metal complex-based drugs in clinical trials have been discussed and deliberated. The pandemic shook the entire world and claimed quite a percentage of the global population. Metal complex-based drugs are already established for their antiviral property with respect to enveloped viruses and extrapolating them for COVID-19 can be an effective way to manipulate drug resistance and mutant issues that the current anti-COVID-19 drugs are facing.

## 1. Introduction

Coronaviridae is a rather unique viral family, which has a significantly large RNA and a distinct morphology, as well as an extraordinary ability to pass from animals to humans. It is to be noted that three of the most transmissible and pathogenic coronaviruses from this family have breached the species barrier and produced lethal outbreaks, and unprecedented health emergencies since the 21st century began. Two human coronaviruses—severe acute respiratory syndrome coronavirus (SARS-CoV-1) and Middle Eastern respiratory syndrome coronavirus (MERS-CoV)—have caused nearly 10,000 cumulative cases with 10% and 34.4% fatality rates, respectively. SARS-CoV-2 originated from Wuhan in December 2019 [1]. Most countries were affected by this pandemic. It has swiftly spread worldwide, infecting almost 22 million people, and resulting in 770,000 fatalities, with a projected mortality rate of 3.6%.

Coronaviruses are classified into four genera: α-, β-, δ- and γ-coronaviruses. Mammals commonly contract moderate respiratory or gastrointestinal infections from Coronavirinae. Most BatCoV (coronavirus isolated from bats) are β-coronaviruses. β-coronaviruses include SARS-CoV-1 and -2 (subgroup 2b), MERS-CoV (subgroup 2c), and HCoV-OC43 and HCoV-HKU1 (subgroup 2a). HCoV-229E and HCoV-NL63 are α-coronaviruses (subgroup 1b) [2,3,4]. SARS-CoV-2 has a 29,891-nucleotide genome that encodes 9860 amino acids, four structural proteins, and sixteen non-structural proteins (NSPs). The 3-chymotrypsin-like protease (3CL-PR), RNA-dependent RNA polymerase (RdRp), and spike protein are potent and selective druggable targets [5,6,7]. NSP12 consists of an N-terminal NiRAN domain, an interface domain, and a C-terminal RdRp domain. It is essential for viral genome replication and mRNA synthesis [8,9,10]. RdRp has the conventional design of viral polymerases [11] with three subdomains: a fingers subdomain (residues Leu366 to Ala581 and Lys621 to Gly679), a palm subdomain (Thr582 to Pro620 and Thr680 to Gln815), and a thumb subdomain (His816 to Phe920) [8]. The palm subdomain’s active site has at least five conserved A-E motifs [8,12]. The catalytic residues Ser759, Asp760, and Asp761 are in motif C, while motif A has the divalent-cation–binding residue Asp618. The only confirmed authenticated treatment for SARS-CoV-2 infection is the nucleotide analogue pro-drug remdesivir, an RdRp inhibitor [13]. After witnessing the havoc COVID-19 caused, researchers worldwide have been frantically investigating many therapeutic options to find candidates that can combat coronavirus outbreaks.

Apart from the nucleotide analogue prodrug remdesivir, no other validated pharmaceutical treatment for SARS-CoV-2 infection or other human pathogenic coronaviruses is available. Remdesivir is the only FDA-approved emergency treatment. Remdesivir has not yet been tested in large-scale coronavirus clinical studies [14,15]. Molnupiravir, which is the first oral medicinal formulation for severe symptoms, has been studied in several clinical investigations [16,17,18]. The FDA and EMA are still investigating this formulation, but the UK has approved it. The first oral antiviral for COVID-19, Lagevrio (molnupiravir), was approved by MHRA. Despite recent headlines, molnupiravir appears to only work in early-stage COVID-19 patients. Its efficacy is minimal in hospitalized patients with advanced disease, which is foreseen as a limitation [19]. Vaccination works best for viral epidemics, as per history [20]. To stop the virus, many governments have accelerated vaccine licensing. SARS-CoV-2 vaccines were given to 22% of the world’s population by mid-June 2021 due to vigorous vaccination programs. The COVID-19 epidemic continues despite widespread public immunizations [21]. So, scientists worldwide are searching for novel SARS-CoV-2-fighting substances. Molecular knowledge of the virus is growing, and various therapeutic targets are being identified and defined. Drug repurposing, which is a system of using medications currently in clinical use for a different therapeutic indication, is a simple way to find active molecules to attack COVID-19. Tocilizumab, Chloroquine, and Remdesivir are interesting candidates for COVID-19 medication identified by drug repurposing. Metal-complex-based drugs may also be effective against COVID-19, according to Messori et al. [22,23]. It is interesting that none of the repurposed metal compounds have been put through clinical testing [24]. 

This review highlights the potential benefits that come from metal-complexes when used as antiviral compounds. The current status of applying metal-complex-based drugs as anti-COVID-19 therapeutics has been surveyed and the challenges and gaps as well as future recommendations exclusively listed and discussed. 

## 2. Metal Complexes in Medical Applications

The various medicinal benefits reported from metal complexes are projected in Figure 1. Metals and metal complexes are being employed more in cancer treatment ever since their discovery [25,26,27,28,29]. Metals bound to N, O, and S form chelate rings that bind the metal more firmly than the non-chelate version. Proteins, enzymes, and DNA are electron-rich, but metal ions are electron-deficient; hence, metal ions interact with several key biological molecules [30]. Metallic complexes can bind large biological molecules better than metal-free organic substances, making them enzyme inhibitors [31]. Metal complexes can coordinate with non-metalloenzymes’ active sites and chelate metalloenzymes. Metals also catalyze reactive oxygen species (ROS) [32]. Main group elements and transition metals have been tested for anticancer properties [33,34,35].

Platinum metal complexes, including cisplatin, carboplatin, and oxaliplatin are among the most active and extensively used cancer chemotherapeutic medicines [36,37,38]. In vivo, platinum compounds crosslink DNA, resulting in programmed cell death [39]. Due to primary and secondary resistance, these medications are only useful for sarcomas, small cell lung cancer (SCLC), ovarian cancer, lymphomas, and germ cell malignancies [40,41]. Hence, research groups create specific platinum complexes for cisplatin-resistant malignancies [42,43,44,45]. Metal coordination changes lipophilicity, which restricts cell entrance, and reduces side effects. Metal complexes have higher biological activity than free ligands [46]. Thiosemicarbazones and their metal complexes are used in medicinal chemistry and may suppress cancer cell activity [47,48]. Triapine (Vion Pharmaceuticals Inc., New Haven, CT, USA) inhibits DNA manufacture in leukemia L1210 cells by blocking ribonucleotide reductase [49].

Platinum-based drugs specifically target head and neck cancers. These heterocyclic thio-semicarbazone derivatives and their platinum and palladium complexes perform many pharmaceutical activities, including anti-tuberculosis [50], antibacterial [51], antitumor [52], antiprotozoal [53], antimalarial [54], antimicrobial [55], antiviral [56], antifungal [57], anticonvulsant [58], and anti-trypanosomal [59,60]. The metal complexes 5-substituted thiophene-2-carboxaldehydes yield thio-semicarbazones and their platinum(II) and palladium(II) complexes. In vitro antiviral and cytostatic/toxic actions of ligands and platinum and palladium complexes have been assessed. NAMI-A and KP1019 are potential ruthenium-based antitumor drugs [61]. These are a few notable mentions of the various biological applications, especially medical applications, enabled successfully by metal complexes. Figure 2 summarizes the various chemical properties of metal complexes that give them their biological attributes.

## 3. Metal Complexes for Antiviral Applications 

Metallodrugs are becoming increasingly therapeutic. In this section, we briefly review the authenticated antiviral reports from various published sources. The most popular are the gold-associated complexes. Auranofin has been found to be highly effective against HIV, as validated by pilot clinical trials [62]. Sanarino and Shytaj [63] found that auranofin treats HIV better than chloroquine. With respect to biological systems [64], drug complexation with metal ions generally improves their antiviral activity [65].

There are various reports on metal ion complexation of antiviral medications like valacyclovir, acylhydrazones, ribavirin, 2-hydroxybenzamides-based compounds that have been successfully used to treat influenza virus, and adefovir, biguanide derivatives used to treat herpes, as well as cidofovir which has been used to cure cytomegalovirus retinitis, and polyoxovanadates. Specifically, protonation constants of Herpes simplex virus (HSV)-fighting valacyclovir (l-valine, 2-[(2-amino-1,6-dihydro-6-oxo-9*H*-purin-9-yl)methoxy]ethyl ester) were reported. Three 2-hydroxy-3-methoxyphenyl acylhydrazones (HL1, HL2, and H2L3) that inhibit HSV and vaccinia virus via acid-base and coordinating action towards Cu(II), Mn(II), and Mg(II) [66] have been studied. Ribavirin (1,β-d-ribofuranosyl-1,2,4-triazole-3-carboxamide, RBV) complexation against HCV was also examined. RBV-Cu(II) complexes, whose hepatic and serum concentrations rise due to HCV infection, were examined [67]. Various complexes of biguanide derivatives NRR’(NH)CNHC(NH)NH_2_ with Co(II) or Co(III) ions could inhibit herpes virus 9 (type 2) [68]. The solvent impacts on ligand dissociation constants increased with ethyl alcohol content. PMEA fights viruses like herpes and HPMPC treats AIDS-related CMV retinitis [69].

New antiviral vaccines have been optimized for efficacy and long-term immune response due to increased knowledge of the immune response mechanism [70]. Due to their chemical characteristics and immune system stimulation, metal complexes are considered promising when it comes to vaccination therapy. Mohamed et al. [71] used tenoxicam-based ternary complexes (H2Ten) chelated with biologically relevant metal cations Co(II), Mn(II), Zn(II), Cu(II), Ni(II), Fe(III), and Cr(III) to formulate the Infectious Bovine Rhinotracheitis (IBR) vaccine. Sonbati et al. developed polymeric complexes based on 5,5′-[3-diyl)]bis(quinolin-8-ole) (H2L) with Cu(II), Zn(II), and Cr(III), Co(II), Ni(II), as BRS vaccines [72]. BRS vaccine attenuated with polymeric complexes had no negative effects and provoked enhanced immune responses (from 2-week treatment onwards).

Due to their cell attraction, cytokine generation, and humoral immune response, metallic nanoparticles (MeNPs) are attractive adjuvants [73,74,75,76,77]. In recent years, metallic nanoparticles, particularly gold nanoparticles (AuNPs), have been good nanovaccine candidates because of their low toxicity, chemical inertness, biocompatibility, and biodistribution. Sengupta et al. [78] recently reviewed the use of AuNPs in nanovaccines against several infectious agents, including viruses. Teng et al. [79] developed FMD vaccinations using AuNP adjuvants. AuNPs and FMD virus-like particles (VLPs) as vaccine antigens have increased macrophage activation and immunological response. Table 1 lists the various metal complexes that have been reported for their antiviral applications. 

### Nano-Based Metal Complexes

Nanotechnology has enabled drug-delivery nanosystems. Metal-based nanomaterials have been employed as drug vehicles because of their exceptional physical and chemical properties. To name a few, their properties include greater control of drug release at the action site and specificity, increased drug absorption, and decreased side effects [96,97,98,99]. Gold and silver nanoparticles have been widely employed to control the release of antiviral medicines [100]. Gold nanoparticles (AuNP) are attractive due to their nano sizes, inertness, and multivalence, which facilitate drug conjugation and simultaneous transport [100]. Garrido et al. [89] examined the blood-brain barrier-crossing capabilities and antiviral activity of gold nanoparticles coupled to raltegravir (RAL) molecules for HIV treatment. RAL was thiol-functionalized to increase drug conjugation with AuNP. Four molecules of RAL with AuNP reduced HIV-1 replication by 25% after five days, but increasing conjugated RAL molecules decreased the antiviral effect. RAL’s inclusion in the metal core gives the AuNP-RAL combination its antiviral action. AuNP alone did not demonstrate the above effects.

Nanocarrier formulation aims to create non-toxic systems that improve therapeutic efficacy and reduce toxicity. Horcajada et al. [93] created Iron(III)-based MOFs as nanocarriers to improve AIDS medication delivery. Porous iron carboxylate nanoMOFs’ non-toxicity, biocompatibility, and adaptability as sponges that trap pharmaceuticals of varied polarity, have been useful. AuNPs’ characteristics and transversal powers make them popular vaccination carriers. Chen et al. [90] tested AuNPs as FMD peptide vaccination vectors. A synthetic peptide-like FMD viral protein was coupled to 2, 5, 8, 12, 17, 37, and 50 nm AuNPs. Protein conjugation to AuNPs was maximized by C-terminal cysteine functionalization. pFMDV-AuNPs in the 8–17 nm range produced three times stronger immune responses than pFMDV-keyhole limpet hemocyanin (pFMDV-KLH). AuNPs alone showed no antibody response. Yan et al. [94] produced nickel-coated lipid nanoparticles to transmit the HIV virus’s His-Tat cationic antigen. His-Tat/Ni-NPs had less toxicity than dendritic cells but failed to improve adjuvant activity compared to the anionic NPs / Tat /sodium dodecyl sulphate (SDS/NP) complex. p24 and Nef co-transport was examined in Ni-NPs. This formulation is excellent for multivalent vaccines since female BALB/c mice had a stronger immunological response. Zazo et al. [91] used AuNPs to deliver HIV antiviral stavudine to primary macrophages. AuNPs of 10–70 nm were functionalized with PEG, PEI, and citrate and coupled to stavudine. The scientists found that AuNP-stavudine increased macrophage morphological diversity and drug efficacy compared to stavudine alone. It took substantial amounts of stavudine alone to achieve the same outcomes, demonstrating that the carrier increases intracellular absorption. The authors speculated that stavudine’s hydrophilic characteristics impede passive absorption.

Paul et al. [92] produced AuNPs with siRNA to promote metabolic stability for DENV therapy. Encapsulating AuNPs complexes in a cationic polymeric bilayer increased siRNA-AuNPs conjugation. The generated cationic compound increased its capacity to pass the negatively charged cell membrane, reducing DENV-2 multiplication. Bibi et al. [101] suggested doping fullerene C_60_ with metals (Fe, Cr, and Ni) to deliver favipiravir (FPV), a commonly used anti-infection medicine. The authors found that doping C_60_ with Fe, Ni, or Cr increased the rate of FPV administration, led to controlled release, and reduced side effects.

Zachar [95] suggested using colloidal metal nanoparticles (NpC) to deliver medications via aerosols for respiratory tract infections. NpCs, notably 2–10 nm sized silver NpCs, with a strong negative zeta potential, engage electrostatically with the spike proteins of normally positively charged viruses, leading to improved effectiveness. Niikura et al. [85] synthesized gold nanoparticles coated with WNV envelope (E) protein as adjuvants. Fischer et al. researched a way to boost WNV vaccination efficacy [102]. They constructed nanolipoprotein particles (NiNLPs) from phospholipid and lipoprotein bilayers with lipid-heads that chelated nickel ions. NiNLP immobilized a polyhistidine-functionalized WNV envelope protein (trE-His). NiNLP-trE-His injection provided a stronger immune response over trE-His alone.

## 4. Metal Complexes and Anti-COVID-19 Applications

Several chemicals are being tested, but the optimal treatment for COVID-19 is yet to emerge. Nevertheless, academic scientists worked with pharmaceutical companies to develop a variety of vaccines in 2020. These vaccinations protect healthy people but may not protect viral variations or immunocompromised patients. Hence, COVID-19 requires innovative treatments [103]. Researchers are developing and screening metal complexes to find novel therapeutics. Drug metal complexes apart from treating COVID-19 can additionally also provide other benefits. For instance, the body needs metal ions to make haemoglobin, zinc fingers regulate gene activities and recognize DNA [104,105], and zinc and copper are needed for growth and immune system development. Auranofin, a gold medication, may be a possibility for drug repurposing, although few other licensed metal complexes exist. 

Computational approaches to assess chemical compound activity based on drug and target structure were deployed on vanadium compounds against COVID-19 [106]. Copper complexes containing chloroquine and hydroxychloroquine contain balanced polar and non-polar groups compared to the parent medication. Its structure allows metallodrug binding to ADP-ribose-1 monophosphatase enzyme’s active site of the virus. Inhibiting this enzyme prevents COVID-19 [107]. Another group reported in silico synthesis of Cu (II) and Co (II) thiazole-based ligand complexes and detailed their role as COVID-19 protein interactors using molecular docking [108]. Refat et al. validated the synthesis of binuclear Schiff base complexes (Zn[II], Cu[II], Co[II], and Ni[II]), and their biological activity was tested using Molecular docking methods. Ni(II) bound COVID-19 protease (6LU7) better in molecular docking experiments [109]. Rad et al. also presented a new COVID-19 treatment using simulations of Fe, Cr, and Ni transition metal-doped fullerenes–favipiravir complexes [110]. In another study, gibberellic acid (HGA), a plant hormone, was used to generate complexes with transition metals Cr(III), Ni(II), Hg(II), Zn(II), Co(II), Cd(II), Mn(II), and Cr(III). These new compounds were computationally tested for interaction with the COVID-19 active site 6LU7. Mn(II) had increased binding energy with the active site and was a putative 6LU7 inhibitor, representing probable anti-COVID-19 activity [111]. Omar and Ahamed synthesized tridentate Schiff base metal complexes of Zn(II), Cu(II), Mn(II), Ni(II), Fe(III), Cd(II), and Cr(II) (III) and screened for anti-COVID-19 activity using MOE2008 molecular docking experiments. They investigated the binding mechanisms of many metal complexes against COVID-19 major protease (SARS-CoV-2) combined with the inhibitor UAW247 (PDB ID: 6XBH). Cr(III) complex demonstrated low binding energy and stable interaction, pointing out strong COVID-19 antiviral activity [112].

Zn(II) boosts immunity and inhibits viral RNA-dependent RNA polymerase. Intracellular Zn(II) levels have been shown to influence DNA and RNA viruses, notably respiratory viruses like influenza, picornaviruses, and respiratory syncytial virus. Chloroquine and hydroxychloroquine work by introducing additional Zn(II) into cells as ionophores. Consequently, Zn(II) may directly block the SARS-CoV-2 replicative cycle [113]. Poupaert et al. also endorse Zn(II) for COVID-19 therapy. Quantum mechanics molecular simulations pointed out molecular Zn^++^ interactions. First- and second-generation macrolides like azithromycin (Zn^++^-antibiotic complex) apparently possess anti COVID-19 potential [114].

Despite metallodrugs’ growing therapeutic value, no meaningful measures have been taken to restrict their pandemic effectiveness. A few noteworthy initiatives exist. Marzo and Messori were the first to try metal-based pandemic medications. Auranofin was proposed against SARS-CoV-2 [115]. The researchers exploited the positive attributes of this compound, namely its biocompatibility (tolerance to the biological system), low toxicity, and multi-targeting. Two more experiments strengthened Marzo and Messori’s idea, Auranofin at low micromolar concentrations suppressed the virus by 95% after 48 h in human cells. It drastically inhibited human cell cytokine expression [116]. Gil-Moles and colleagues found that auranofin inhibited the spike protein’s interaction with ACE2’s active site. This is the virus’s main entry point into human cells. The chemical strongly inhibited SARS-CoV-1 and SARS-papain-like CoV-2’s protein (PLpro), a viral replication enzyme. This enzyme is the first to block this target protein experimentally [117].

Bismuth complexes have also been reported for their anti-COVID-19 potential. In 2007, researchers evaluated bismuth complexes’ capacity to attach to the 100-residue cysteine-rich metal-binding domain (NTPase/helicase), which plays a crucial role in SARS-CoV-2 life cycle [118]. Bismuth compounds inhibited SCV helicase enzymatic activity, paving the way for SARS-CoV-2 studies. In a 2020 study by Yuan et al. [119], ranitidine bismuth citrate was tested against COVID-19, in vitro and in vivo. Cell cultures and animals exhibited efficacy. In cell cultures, this combination protects against SARS-CoV-2 with low toxicity. This metal complex significantly decreases virus proliferation in Syrian hamsters, lowering the respiratory viral burden. Researchers also noted that the combination inhibited viral helicases in vitro, confirming prior studies on the enzyme’s usefulness as a pharmaceutical target [120] and bismuth complexes’ capacity to target it. However, these results have not been put through clinical trials. The one exception is a case study reported in the American Journal of Gastroenterology that indicated that a patient with exacerbated Crohn’s disease due to SARS-CoV-2 infection, who showed significant improvement [121]. His doctors prescribed bismuth subsalicylate 525 mg orally 2–4 times a day. Over six days, this treatment reduced diarrhoea, cough, appetite, energy, and virus-triggered symptoms. Figure 3 portrays the three-faceted anti-COVID-19 contribution that is made possible by metal-complexes.

### Mechanism of Anti-COVID-19 Activity of Metal Complexes

Modern medicine has now paved the way to accommodate metallodrugs. The existing modes of action of pharmacological compounds with antibacterial and possible antiviral activity of metal complexes are discussed in this section. In a January 2020 review [122], researchers attempted to categorize metal complex drugs based on their action modes. Here we discuss mechanisms that we think are relevant to the anti-COVID-19 perspective, at this point. The first mechanism involves the covalent bonding of metallodrugs to biological targets [123]. Cisplatin and auranofin fall within this mechanistic category. These two medicinal compounds affect stereochemistry through their biological targets’ covalent connections. Biological processes are thus inhibited [124]. Auranofin blocks cell death oxidoreductive processes, activates the unfolded protein response (UPR), induces endoplasmic reticulum (ER) stress, and inhibits thioredoxin redox enzymes [125]. Thioredoxin transfers electrons to oxidative stress-protecting enzymes. Auranofin inhibits thioredoxin reductase by dislodging the gold complex’s ligands and forming a covalent connection with the active center’s cysteine residues [126]. Auranofin induces apoptosis by blocking these enzymes. Auranofin reduces cytokines and stimulates cellular immunity [127]. ER oxidative stress and UPR activation are responsible for coronavirus virulence, making this mechanism of action useful against coronaviruses [128]. According to the study, cells infected with this family of viruses may overexpress spike proteins. Other viral proteins activate UPR [129]. Inhibiting thioredoxin reductase and other redox enzymes may impair SARS-CoV-2 protein production [130]. Auranofin’s anti-inflammatory properties make it a potential COVID-19 drug. Systematic observation has shown that SARS-CoV-2 infection causes immediate respiratory inflammation and a cytokine storm with interleukin-6 (IL-6) upregulation [131]. Auranofin suppresses IL-6 signalling by inhibiting JAK1 and STAT3 phosphorylation, according to Nam-Hoon Kim and colleagues [132]. Inhibiting the inflammatory condition could save lives and lessen their virus-related mortality. 

The second mechanism of action of metal complexes against viruses, especially against SARS-CoV-2, is that of redox-active metal centers. Many metals or metal ions can be oxidized. Their oxidation states alter substituent dynamics and environmental chemical and biological processes [133,134]. Previous research has shown that the oxidation state and other intracellular variables affect virus life cycles [135]. Being intracellular parasites, viruses have several ways to exploit and damage cells [136]. Consequently, infected viruses shift the redox balance towards oxidation [137]. Respiratory viruses overproduce reactive oxygen species (ROS) and decrease glutathione, the cell’s main antioxidant. ROS and glutathione decrease and enhance viral replication pathways [138]. NOX oxidase, which has seven members, overproduces ROS during infection, with NOX2 being crucial to viral proliferation. Its absence reduces respiratory infection duration and severity [139]. NOX4 is an even better target. Lung epithelial cells overexpress NOX4 and produce ROS following infection [140]. ACE2, which is the principal receptor coronaviruses employ to enter cells, regulates NOX4 ROS generation [21]. ROS generation activates MAPKs and facilitates nuclear extraction of the filial ribonucleoprotein, allowing viral replication [141]. Thus, with SARS-CoV-2, overexpression of ROS and NOX2 has been linked. Damiano and her colleagues suggest reducing cell oxidative stress to reduce COVID-19 infection and safeguard high-risk patients [142]. In this direction metal complex-based drugs hold promise; however, their potential in this direction has not been proven.

The third mechanism is via photodynamic treatment (PDT) and photoactivated metal drug complexes. Due to their unique characteristics, metal complexes are ideal for such therapies [143]. Certain metals can absorb visible light and diatomically absorb two photons in the near-infrared region. Metals boost spin–orbit coupling, allowing the triplet state that produces simple oxygen. Finally, metal-containing compounds in PDT therapy are photostable. This mechanism is largely used effectively in treating cancer. Yet, it can also be applied to treat viruses [144]. PDT was first used against herpes genitalis in the early 1970s [145]. Since then, this approach has been tested against HPV, HIV, CMV, and others [146,147]. Enveloped viruses responded better to PDT, according to research reports. SARS-CoV-2, being an enveloped virus, was also envisioned to respond well [148,149]. This approach activates photosensitizers (PS) to produce ROS. PSs activate by absorbing photons at a specific wavelength. After a few nanoseconds of electron transport, the PS enters the triple state. ROS generation is the last stage before entering one of the two photochemical routes. Oxidative stress from ROS generation damages nucleic acids and microbial systems. The approach is advantageous since PS are non-toxic chemicals that only become toxic at the target spot following exposure to a specific wavelength of radiation [150]. To our knowledge, no clinical investigation has tested this approach for COVID-19 infection. We found important in vitro research evidence that suggests the strategy may work [151,152]. Nonetheless, it has been tested against such other coronaviruses as MERS-CoV. [153]. PDT’s ROS generation can inactivate SARS-structural CoV-2’s proteins, based on its structure. Photochemical inactivators, including riboflavin, curcumin, and chlorophyll derivatives inhibit coronaviruses, according to prior studies [154]. Mechanistically and administratively, this technique is interesting. Nebulization with a catheter for light supply is one way to treat SARS-CoV-2, which affects nasal and oropharyngeal epithelial cells [155,156]. In 2020, Dias suggested using this method to treat COVID-19 adjunctively [157]. Several researchers endorse this SARS-CoV-2 treatment despite its specificity and limited use against viral diseases [158]. 

SARS-high CoV-2’s mutation rate makes traditional antiviral medication design problematic. This virus modifies its protein targets often, leading to resistance to these drugs [159]. On the other hand, metal-based drugs do not act by interacting with specific molecular targets of the virus. Their activity lies in the virus’s general environment, an advantage that helps them to be less vulnerable to viral mutations and thereby to become a better pharmaceutical approach compared with the other options.

## 5. Challenges, Future Perspectives, and Conclusions

Metals and metal complexes, endowed with “magical” qualities, have long fascinated physicians and played a crucial part in contemporary pharmacology since the late 19th/early 20th century. Gold, bismuth, antimony, and mercury compounds have been successfully applied to treat infectious disorders such as tuberculosis, syphilis, and parasitic diseases [160,161]. Even arsenicals were often used in clinics. Due to legitimate concerns on their systemic toxicity and the arrival of new organic medications with greater pharmacological performance and reduced toxicity, these inorganic compounds were eventually abandoned. Yet, some inorganic medicines are still in clinical practice for a few particular applications where they provide valuable and irreplaceable efficacy with acceptable toxicity [162]. In spite of their systemic toxicity, platinum medications are still applied to nearly 50% of cancer chemotherapy procedures [163]. Given the seriousness of cancer, their toxicity may be acceptable. Nonetheless, antimony compounds for leishmaniasis, arsenic trioxide for promyelocytic leukemia, and bismuth compounds for *Helicobacter pylori* infections are noteworthy [163]. For the last four decades, interest in metal-based medications, because of the clinical success of cisplatin (first approved by the FDA in 1978), provoked and caught the attention of the international scientific community working in inorganic medicinal chemistry. The fact behind the spurred enthusiasm for metal-complex drugs is that they contain a wide variety of metal centers with unique chemical and reactivity properties arising from the metal’s electronic structure, coordination sphere, ligands, oxidation state, redox potential, etc. These chemical features cannot be reproduced by simple organic compounds. Metal complexes may directly inhibit enzymes, modify transcription factors, interact with a range of biomolecules through coordinative bonding, boost lipophilicity, affect cell membrane activities, interfere with the cell cycle, and more.

Cirri et al. [22] have proposed two strategies for identifying anti-COVID-19 metal complex-based drugs. The first strategy is through drug discovery from clinically proven metal-based medicine by repurposing clinically licensed metal-based medications for COVID-19 treatment. This is a simple way to find effective metallodrugs. The second strategy is through screening metal-based chemical libraries for drugs We recommend rigorous in vitro testing of vast and representative libraries of metal-based medicinal compounds against SARS-CoV-2 replication. Families of medicinally suitable metal-based medicines with tolerable toxicity should dominate investigational panels. Bismuth, ruthenium, and antimony compounds may be ideal due to their low toxicity. 

In the course of this review, we identified a gap in the usage of the new generation state-of-the-art materials, such as fullerenes, graphene, and carbon nanotubes in the making of metal complexes. A recent study [149] based on the principle that singlet oxygen-generating compounds inactivate encapsulated viruses via photodynamic processes, demonstrated the use of Buckminsterfullerene (C_60_), a water-insoluble photosensitizer, to inactivate encapsulated viruses. VSV and Semliki Forest virus (SFV, Toga6iridae) were studied in this context. C_60_ produces much singlet oxygen and is photo-oxidatively inert. It is also easy to recycle from aqueous solutions. Hence, it may help to inactivate viruses in biological systems. Carbon materials have a lot to offer and hybridizing these with traditional metal complexes could greatly benefit this cause. The metal-complex aspect also lacks the nanofactor; few of these metal complexes involve nanometals. Nanotechnology had breached and broken through innumerable limitations of bulk materials. There is no doubt that more nanoaspects need to be incorporated into anti-COVID-19 prospects being sought from metal complex-based drugs. 

Photodynamic therapy is one application that has greatly benefitted from metal complexes; yet, strangely, little has been done in this direction with respect to COVID-19 treatment. This is another gap that needs to be bridged. The other essential, crucial aspect lacking in the theme under discussion is that of clinical trials and testing. Mostly simulated/computational/in silico studies (with respect to anti-COVID-19 metal-complex applications) are reported. Few are done in vitro, and fewer still in vivo; clinical trials are almost nonexistent. Clinical trials are the actual green signal for any prescribed drug or treatment option. 

The anti-COVID-19 options from metal complexes have been explored, and the current scenario with respect to the status quo of metallo drugs available and tested against SARS-CoV-2 has been reviewed and presented. The need for introducing and hybridizing conventional metal complexes with more new-age nanomaterials and the mandate for clinical trials for testing of the metallo complexes have been emphasized. 

## Figures and Tables

**Figure 1 molecules-28-03354-f001:**
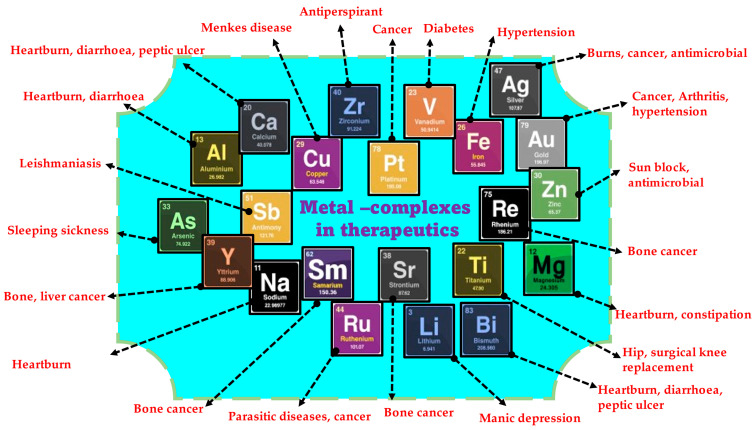
Validated medicinal properties of various metal/metal complexes.

**Figure 2 molecules-28-03354-f002:**
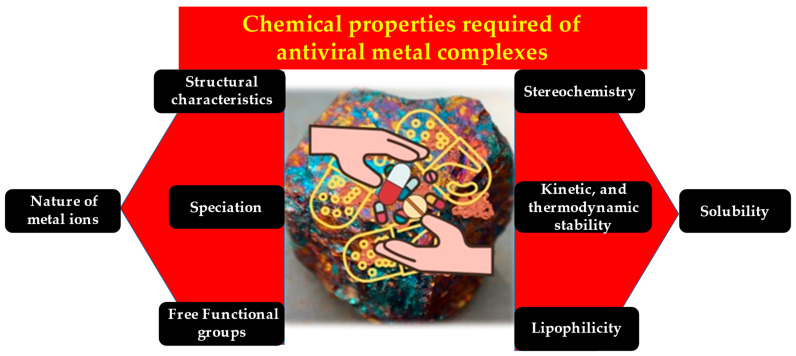
The unique chemical characteristics of metal complexes add up to their antiviral property.

**Figure 3 molecules-28-03354-f003:**
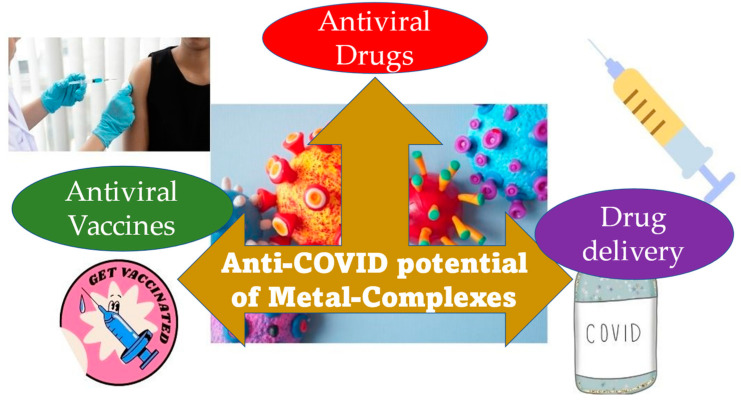
Three modes of contribution from metal complexes towards the anti-COVID-19 activity.

**Table 1 molecules-28-03354-t001:** Comprehensive list of antiviral metal-complexes.

Metal-Complex	Virus	Application	Reference
Au(I)–2,3,4,6-tetra-*O*-acetyl-l-thio-β-d-glycol-pyranoses-*S*-(triethyl-phosphine) (AF)	HIV-1, SARS-CoV-2	Antiviral therapy	[23,80,81]
Au(I)–Aurothiomalate	HIV-1	Antiviral therapy	[80]
Au(I)–Aurothioglucose	HIV-1	Antiviral therapy	[82]
Ga(III)–Curcumin	HSV-1	Antiviral therapy	[83]
Co(III)–Doxovir	HSV-1	Antiviral therapy	[84]
Cu(II) Ni(II) Co(II) Zn(II) Cr(III)–Tenoxicam, valine	Infectious Bovine Rhinotracheitis (IBR)	Vaccine therapy	[71]
Cu(II) Ni(II) Co(II) Zn(II) Cr(III)–5,5-[3-diyl)]bis(quinolin-8-ole)	Bovine respiratory syncytial (BRS)	Vaccine therapy	[73]
AuNPs–FMD virus-like particles (VLPs)	Foot-and-mouth disease (FMD)	Vaccine therapy	[79]
AuNPs–West Nile Virus (WNV) envelope (E) protein	West Nile Virus (WNV)	Vaccine therapy	[85]
AuNPs–STG antigen	Enteropathogenic coronavirus of transmissible porcine gastroenteritis (STG)	Vaccine therapy	[86]
AuNPs–PEG-S461-493	SARS-CoV-2	Vaccine therapy	[87]
AuNPs–Protein S	SARS-CoV-2	Vaccine therapy	[88]
AuNPs–Raltegravir (RAL)	HIV	Drug delivery	[89]
AuNPs–FMD virus protein	FMD	Drug delivery	[90]
AuNPs–Stavudine	HIV	Drug delivery	[91]
AuNPs–siRNA	DENV	Drug delivery	[92]
Fe(III)–NanoMOFs [59]	HIV	Drug delivery	[93]
Fe(III)–Tenoxicam, valine	IBR	Vaccine therapy	[71]
NiNPs–His-Tat cationic antigen	HIV	Drug delivery	[94]
AgNPs–Spike proteins	SARS-CoV-2	Drug delivery	[95]

## Data Availability

Not applicable.

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
