# Peer review of "A Comprehensive Survey on the Expediated Anti-COVID-19 Options Enabled by Metal Complexes—Tasks and Trials"

_molecules, 2023, doi:10.3390/molecules28083354_

Round 1

Reviewer 1 Report

This very interesting review treats of the varied antiviral benefits that could be harnessed through use of metal complexes. For this reason the antiviral properties of metal complex-based drugs against enveloped viruses as the COVID-19 are illustrated and are summarized the anti-COVID-19 results deriving from the exploitation of the pharmacological activity of metal complexes. Finally the need to put to test metal complex-based drugs, even in clinical trials, is discussed and suggested as a future challenge. The conclusion is that this approach could be an effective alternative to the current anti COVID-19 drugs that frequently collide with resistance and mutants issues.

I have a concern about the english language that should be cheked. Moreover, another concern is about the literature cited on the role of metal compounds in the development of new antiviral drugs, which should be better checked (for instance the paper https://doi.org/10.1016/j.ccr.2021.214276 is missing).

Author Response

This very interesting review treats of the varied antiviral benefits that could be harnessed through use of metal complexes. For this reason the antiviral properties of metal complex-based drugs against enveloped viruses as the COVID-19 are illustrated and are summarized the anti-COVID-19 results deriving from the exploitation of the pharmacological activity of metal complexes. Finally the need to put to test metal complex-based drugs, even in clinical trials, is discussed and suggested as a future challenge. The conclusion is that this approach could be an effective alternative to the current anti COVID-19 drugs that frequently collide with resistance and mutants issues.

Ans. We thank our reviewer for the nice words, very encouraged. We will execute your below cited suggestions in our revised version. Thank you for your efforts and time. 

I have a concern about the english language that should be cheked. Moreover, another concern is about the literature cited on the role of metal compounds in the development of new antiviral drugs, which should be better checked (for instance the paper https://doi.org/10.1016/j.ccr.2021.214276 is missing).

Ans. We apologize for the language issues, we have now had the manuscript thoroughly read and edited by a native speaker, the manuscript has been rid of language issues. thank you. Also the references we have cross checked and updated. thank you very much, 

Reviewer 2 Report

MS ID: molecules-2288031

Author(s): Judy Gopal, Manikandan Muthu and Iyyakkannu Sivanesan*

Title: A Comprehensive Survey on the Expediated Anti-COVID-19. Options Enabled by Metal Complexes – Tasks and Trials

Recommendation: not publishable at this stage. It would be in the interface between “reject” (mainly) and to reconsider after “several and crucial major revision”.

The present review wanted to focus on the state-of-art of the antiviral (anti-COVI-19) potential of metal complexes.

As stated from the authors, the anti-COVID-19 perspective from exploiting the pharmacological aspects of metal complexes have been summarized. Specifically, they argued about possible challenges and gaps in this therapeutic area, by focusing on the nanoaspects in metal complexes as well as on the metal complex-based drugs, and their clinical potential. Moreover, the authors claimed that metal complex-based drugs are already established for their antiviral properties with respect to enveloped viruses, thus extrapolating them towards COVID-19 can be an effective approach to overcome drug resistance of the current anti-COVID-19 drugs.

The present review is poorly written and confusely presented. Although the topic is interesting, there are crucial comments outlined below which prevent the publication of this manuscript at this stage. These remarks could be addressed to deeply restructure the manuscript before a possible re-submission for publication. 

In summary, the report here presented is interesting about the topic but it is not satisfactory in terms of general organization and technical quality. The manuscript should be clarified about the rational, the structure and the clarity of presentation. There are crucial points, which are listed below, that should be improved to be considered for a potential future publication in the Molecules.

Major points

-        This is one critical point. Abstract. The abovementioned notes (i.e. “… possibile challenges and gaps in COVID-19 therapeutic area, by focusing on the nanoaspects in metal complexes as well as on the metal complex-based drugs, and their clinical potential” as well as “that metal complex-based drugs are already established for their antiviral property with respect to enveloped viruses, extrapolating them towards COVID-19 can be an effective way to manipulate drug resistance and mutant issues that the current antiCOVID-19 drugs are facing.” should be better and deeply expanded and discussed in the manuscript. The authors claimed useful inputs about the potential of metal-complex based compounds on the management of COVID-19 but I did not find a concrete correspondence with what discussed in the text. Furthermore, the review highlights the potential benefits that come from metal-complexes when used as antiviral compounds, but not exhaustive details and related inputs were reported.

-        The author mentioned other previously published reviews which disclosed about metal-based compounds and SARS-CoV-2 but they must clarify about the originality of the present manuscript, also detailing the differences with respect to such papers. I suggest to add few updated references about this point.

-        Introduction section and “metals complexes in medical applications”. The authors satisfactorily reported a coherent state-of-art of metal/metal complex-based drugs for their applications in several pharmacological and clinical contexts. This section could fit with the scope of the review, but the next chapters appeared not satisfactorily structured.

-        Metal Complexes for Antiviral and anti-COVID-19 applications. Another crucial point: the manuscript is an informative review but the authors should consider to provide a most critical overview of limitations and potential facing metal-based drugs to COVID-19, as mentioned in the title and abstract. As an example, this note can be exemplified in the following concern: Figure 2 summarizes the various chemical properties of metal complexes that give them their biological attributes.”; however, the footnote indicates “The unique chemical characteristics of metal complexes that add up to their antiviral property.” Why and how the indicated characteristics of metal complexes are crucial and specific to provide/improve their antiviral properties? The mentioned characteristics appeared too much generics and did not give any important inputs about the design/development of metal-complex antiviral(anti-COVID-19) drug prototypes. Otherwise, the authors can restructure the review by focusing to a more larger therapeutical context.

-        The authors should focus on the most performing class of compounds. In fact they mixed metal complexes, metallic (i.e. gold) nanoparticles, nanovaccines, putative compounds used for photodynamic therapy etc., without any explanation about characteristics and/or mechanism of actions (just for few of them), exploitation of putative targets, and their interface with COVID-19. Moreover, inputs from metal complexes derived from molecular docking appeared speculative.

-        An authorative statements as to the future of this important issue must be provided. Some important deficiencies must be flagged out with possible solutions.

Minor comments

-        The authors wrote: “Molnupiravir, which is the first oral medicinal formulation for severe symptoms, has been studied in several clinical investigations [16–18]. The FDA and EMA are still investigating this formulation, but the UK has approved it. The first oral antiviral for COVID-19, Lagevrio (molnupiravir), was approved by MHRA. Despite recent headlines, molnupiravir appears to only work in early-stage COVID-19 patients. Its efficacy is minimal in hospitalized patients with advanced disease, which is foreseen as a limitation [19].” Please rephrase this sentence in terms of sense and update.

-        Figures are not stylistically well represented. I suggest to restructure them according with the abovementioned suggestions.

-        The manuscript is marred by several typos /spelling errors. A careful grammatical revisions is strongly suggested.

Author Response

The present review wanted to focus on the state-of-art of the antiviral (anti-COVI-19) potential of metal complexes.

As stated from the authors, the anti-COVID-19 perspective from exploiting the pharmacological aspects of metal complexes have been summarized. Specifically, they argued about possible challenges and gaps in this therapeutic area, by focusing on the nanoaspects in metal complexes as well as on the metal complex-based drugs, and their clinical potential. Moreover, the authors claimed that metal complex-based drugs are already established for their antiviral properties with respect to enveloped viruses, thus extrapolating them towards COVID-19 can be an effective approach to overcome drug resistance of the current anti-COVID-19 drugs.

The present review is poorly written and confusely presented. Although the topic is interesting, there are crucial comments outlined below which prevent the publication of this manuscript at this stage. These remarks could be addressed to deeply restructure the manuscript before a possible re-submission for publication. 

In summary, the report here presented is interesting about the topic but it is not satisfactory in terms of general organization and technical quality. The manuscript should be clarified about the rational, the structure and the clarity of presentation. There are crucial points, which are listed below, that should be improved to be considered for a potential future publication in the Molecules.

Ans. Thank you for your comments and elaborate review and valuable suggestions. We greatly appreciate the effort taken. We have worked on the manuscript and edited the manuscript and worked on checking the smooth flow of the review. Thank you.

This is what we had in mind when we set the flow of the manuscript:

  • First we touch on the various key words involved in the title in the Introduction.
  • Then we give a brief overview of metal complexes in medical applications
  • Then we compiled and presented the antiviral reports of metal complexes
  • The next section we presented a executive summary of all the reports involving metal complexes and anti-COVID-19 applications
  • Finally, in the final section we highlight the gaps, lapses, future direction and conclusions.

So saying, the organization we feel is fine, but we do agree that maybe there were some flaws here and there which may have hampered. We have now sorted things out. Thank you.

Major points

-        This is one critical point. Abstract. The above mentioned notes (i.e. “… possible challenges and gaps in COVID-19 therapeutic area, by focusing on the nanoaspects in metal complexes as well as on the metal complex-based drugs, and their clinical potential” as well as “that metal complex-based drugs are already established for their antiviral property with respect to enveloped viruses, extrapolating them towards COVID-19 can be an effective way to manipulate drug resistance and mutant issues that the current antiCOVID-19 drugs are facing.” should be better and deeply expanded and discussed in the manuscript. The authors claimed useful inputs about the potential of metal-complex based compounds on the management of COVID-19 but I did not find a concrete correspondence with what discussed in the text. Furthermore, the review highlights the potential benefits that come from metal-complexes when used as antiviral compounds, but not exhaustive details and related inputs were reported.

Ans. The entire review focused on compiling the scattered reports on antiviral and anti-COVID-19. Once compiled what was lacking was identified and highlighted. As the final section states, there is not much information regarding the antiCOVID application of metal complexes. What we speculate and project are based on the limited information available. We have improved the final section to discuss whatever could be possibly elaborated. Thank you.

-        The author mentioned other previously published reviews which disclosed about metal-based compounds and SARS-CoV-2 but they must clarify about the originality of the present manuscript, also detailing the differences with respect to such papers. I suggest to add few updated references about this point.

-        Introduction section and “metals complexes in medical applications”. The authors satisfactorily reported a coherent state-of-art of metal/metal complex-based drugs for their applications in several pharmacological and clinical contexts. This section could fit with the scope of the review, but the next chapters appeared not satisfactorily structured.

Ans. The originality of this manuscript is that metal complexes in medical applications and antiviral applications has been reviewed, but with respect to anti-COVID 19, this review has contributed higher than any of the previous reviews. We have structured the sections, better. Thank you.

-        Metal Complexes for Antiviral and anti-COVID-19 applications. Another crucial point: the manuscript is an informative review but the authors should consider to provide a most critical overview of limitations and potential facing metal-based drugs to COVID-19, as mentioned in the title and abstract. As an example, this note can be exemplified in the following concern: Figure 2 summarizes the various chemical properties of metal complexes that give them their biological attributes.”; however, the footnote indicates “The unique chemical characteristics of metal complexes that add up to their antiviral property.” Why and how the indicated characteristics of metal complexes are crucial and specific to provide/improve their antiviral properties? The mentioned characteristics appeared too much generics and did not give any important inputs about the design/development of metal-complex antiviral(anti-COVID-19) drug prototypes. Otherwise, the authors can restructure the review by focusing to a more larger therapeutical context.

Ans. We apologize about the fig 2 footnote, we have modified it. Wherever possible we have added more information on the characteristics. Thank you. In fact there is no direct correlation in form of a authentic report connecting the characteristics to design/development of metal complexes. We have put this aspect forth in the final section while discussion gaps and future direction. Thank you.

-        The authors should focus on the most performing class of compounds. In fact they mixed metal complexes, metallic (i.e. gold) nanoparticles, nanovaccines, putative compounds used for photodynamic therapy etc., without any explanation about characteristics and/or mechanism of actions (just for few of them), exploitation of putative targets, and their interface with COVID-19. Moreover, inputs from metal complexes derived from molecular docking appeared speculative.

Ans. We have now as much as possible grouped these. As we highlighted in the final section, the real issue is that scattered information with less depth and information are only available, that is why this review prompts more authoritative reports with respect to anti COVID applications. In fact we have mentioned in the final section that most of the existing few reports on metal complexes and COVID-19 are speculative involving docking and simulations, this we have pointed out and emphasized that more wet lab and real time reports are required. Thank you.

-        An authorative statements as to the future of this important issue must be provided. Some important deficiencies must be flagged out with possible solutions.

Ans. We have now elaborated and added on such statements in the final section. But with the given scanty information, authoritative statements cannot be made for anti-CoVID, but we have expanded and worked on this part as much as is possible. Thank you.

Minor comments

-        The authors wrote: “Molnupiravir, which is the first oral medicinal formulation for severe symptoms, has been studied in several clinical investigations [16–18]. The FDA and EMA are still investigating this formulation, but the UK has approved it. The first oral antiviral for COVID-19, Lagevrio (molnupiravir), was approved by MHRA. Despite recent headlines, molnupiravir appears to only work in early-stage COVID-19 patients. Its efficacy is minimal in hospitalized patients with advanced disease, which is foreseen as a limitation [19].” Please rephrase this sentence in terms of sense and update.

Ans. Rephrased.

-        Figures are not stylistically well represented. I suggest to restructure them according with the abovementioned suggestions.

Ans. Revised.

-        The manuscript is marred by several typos /spelling errors. A careful grammatical revisions is strongly suggested.

Ans. Sorry about that we have now thoroughly gone through the entire manuscript and fixed these. Thank you.

Round 2

Reviewer 2 Report

MS ID: molecules-2288031

Author(s): Judy Gopal, Manikandan Muthu and Iyyakkannu Sivanesan* 

Title: A Comprehensive Survey on the Expediated Anti-COVID-19. Options Enabled by Metal Complexes – Tasks and Trials

Recommendation: publishable 

This is a revised version of a review focused on the on the state-of-art of the antiviral (anti-COVID-19) potential of metal complexes. The authors detailed the anti-COVID-19 perspective from exploiting the pharmacological aspects of metal complexes. 

In the preview version, I indicated some remarks which could be addressed to deeply restructure the manuscript before a possible re-submission for publication.  

Although I’m still perplexed in some points (regarding the too much structure of the review), I realized the authors reflected on the remarks, and satisfactorily responded point-by-point to all the suggested recommendations raised by the Reviewers. Specifically, additional comments as well as a stylistic revisions have been performed and implemented, and detailed clarifications have been provided.

In summary, the revised manuscript can be supported for publication in Molecules.